# A Bimodal Emotion Recognition Approach through the Fusion of Electroencephalography and Facial Sequences

**DOI:** 10.3390/diagnostics13050977

**Published:** 2023-03-04

**Authors:** Farah Muhammad, Muhammad Hussain, Hatim Aboalsamh

**Affiliations:** Department of Computer Science, College of Computer Science and Information, King Saud University, Riyadh 11451, Saudi Arabia

**Keywords:** bimodal, electroencephalography, facial video clips, emotion recognition, CNN, feature level fusion, Deep CCA, HCI

## Abstract

In recent years, human–computer interaction (HCI) systems have become increasingly popular. Some of these systems demand particular approaches for discriminating actual emotions through the use of better multimodal methods. In this work, a deep canonical correlation analysis (DCCA) based multimodal emotion recognition method is presented through the fusion of electroencephalography (EEG) and facial video clips. A two-stage framework is implemented, where the first stage extracts relevant features for emotion recognition using a single modality, while the second stage merges the highly correlated features from the two modalities and performs classification. Convolutional neural network (CNN) based Resnet50 and 1D-CNN (1-Dimensional CNN) have been utilized to extract features from facial video clips and EEG modalities, respectively. A DCCA-based approach was used to fuse highly correlated features, and three basic human emotion categories (happy, neutral, and sad) were classified using the SoftMax classifier. The proposed approach was investigated based on the publicly available datasets called MAHNOB-HCI and DEAP. Experimental results revealed an average accuracy of 93.86% and 91.54% on the MAHNOB-HCI and DEAP datasets, respectively. The competitiveness of the proposed framework and the justification for exclusivity in achieving this accuracy were evaluated by comparison with existing work.

## 1. Introduction

An emotion, a multifaceted mental process, reflects human perceptions and plays a significant part in human interactions [1]. Nowadays, there are many human–computer interaction (HCI) applications that require research on emotion recognition [2]. The environment in the HCI system is complex and dynamic. In many cases, it requires coordinating its operations with the respondents; therefore, a framework with emotional intelligence can better adjust in such an environment. The HCI system will become more human-friendly if it is enabled to recognize human emotions quickly and precisely [3]. EEG signals can actively characterize variations in the human brain during emotional activity, and emotion recognition based on these signals has become a popular trend among researchers.

Humans may be unable to express their emotions in some situations, such as when they are hospitalized or have other impairments. For instance, a person with alexithymia is incapable of communicating with others about their emotional state due to emotional blindness. Researchers in the field of neuroscience have looked for a connection between alexithymia and the origins of the two hemispheres as well as deficits in the amygdala [4]. Similarly, bipolar disorder is one of the main causes of disability in the world and is characterized by uncontrollable extreme mood swings from abnormally happy to deeply sad. Depression (also known as major depressive disorder) frequently comes with sleep issues and eating disorders. It may result in suicidal thoughts or behaviors. Depression and bipolar disorder affect three portions of the brain: temporal lobe, prefrontal cortex, and amygdala [5]. Therefore, understanding emotions and discovering alternate means to communicate with such people would be beneficial.

Alexa, Cortana, Siri, and other intelligent personal assistants (IPAs) use natural language processing to engage with people. However, when emotion detection is added to IPAs, effective communication and human-level intelligence are increased. Moreover, the automated process of identification of human emotion has become a popular trend among researchers after the development of HCI and Internet of Things (IoT)-based systems for hospitals, smart homes, and smart cities. However, existing HCI systems, in many cases, have irregularities when interacting with humans [6,7]. To put it another way, the communication content of HCI systems is out of date, and their recognition ability could be improved. Enhancing emotion identification in HCI systems and facilitating quick and dependable computational solutions in these systems are essential for resolving this issue [8]. A better model for emotion recognition is one of the important steps toward the solution. Furthermore, emotions are classified as discrete (happy, sad, or neutral) or dimensional (valence and arousal to describe the emotional scale from calmness to excitement or high/low positivity or negativity) [9,10].

The literature on emotion recognition techniques is divided into approaches based on physiological and non-physiological signals. Physiological signals, out of both, are comparatively less vulnerable to subjective influences and hence depict the true state of human emotions. Consequently, it may be concluded that physiological signals are useful and reliable in identifying humans’ true emotional states. The physiological signals that are widely used to detect human emotions include EEG, electrocardiogram (ECG), eye movements, and others. Several authors have reported good results in recognizing emotions using these signals individually and through the fusion of multiple modalities [11]. Further, out of all physiological signals, the EEG is the most difficult for humans to hide or deceive and contains subject-independent data to represent true human emotions [12]. Several studies have also looked at the substantial relationship between EEG signals and emotional states in humans [13,14]. Zhang et al. [14] investigated the connection of various human emotional states with different brain regions using EEG signals. Hence, emotion recognition models using EEG signals are more accurate and reliable as compared to other physiological signals.

For humans, facial expressions are one of the most important ways to convey emotions. Recognition of facial expression is one of the most prevailing, common, instant, and accurate as compared to other signals. However, facial video clips are easy to manipulate while recording by faking the expressions, resulting in unreliable recognition [13,15]. Physiological signals, such as EEG, carry instantaneous emotional variations more accurately but are vulnerable to noise obtrusion [16]. Conclusively, emotion recognition methods based on multiple modalities can compensate for the shortcomings of unimodal methods and acquire better and more reliable results [17]. This aspect is more consistent with the requirements of up-to-date HCI systems. It is worth exploring the methods of bimodal emotion recognition that are consistent with modern HCI systems. Additionally, each person can react differently under the same emotional condition, and one person can react differently on different occasions too. It is hard to acquire large enough datasets, and consequently, we must explore approaches for HCI systems based on relatively small datasets [18].

Most facial expression-based emotion recognition systems require facial markers or front-facing images, which results in inefficient systems in terms of robustness [11]. Additionally, just expecting the deep learning model to be effective at facial expression recognition is insufficient in the absence of a precise strategy to exclude extraneous data from the facial video clips, which might decrease the model’s effectiveness [19]. Additionally, the majority of EEG-based emotion recognition models have poor performance, which is brought on by some of the EEG data channels that do not contain emotion-related information [20,21]. Additionally, considering all the EEG data channels in a deep learning model is not recommended when it comes to time complexity [11]. Finally, the handcrafted fusion methods such as enumerator and Adaboost fusion proposed by Li et al. [22] are limited in terms of performance and urge the need to explore deep learning methods to develop highly correlated features space from multiple modalities. To prove the significance of the fusion method, Liu et al. [23] considered deep canonical correlation analysis (DCCA) to fuse the hand-crafted features from EEG, ECG, and Eye movements. However, the performance of the method using a single modality is not reported, and the study considered only hand-crafted features.

While managing to keep the necessary information in the facial video clips, we have developed a method to discard the frames in each second that were below a quality threshold. Moreover, this study utilizes a deep learning approach to extract more generalized features instead of hand-crafted features to generate a more robust and accurate model for emotion recognition. This work presents an emotion recognition framework using the fusion of facial video clips and EEG trials to recognize three categories of emotions (happy, sad, and neutral). We have considered distinct convolutional neural network (CNN) models to extract features from facial video clips and EEG trials. After feature extraction, the fusion process is the key to accurate and robust emotion recognition. For that purpose, a DCCA approach was exploited to extract the highly correlated features from facial video clips and EEG feature space. Afterward, these highly correlated fused features were used for classification. In separate experiments, facial video clips and EEG features were also individually used for the classification of emotions. Finally, it was concluded that the feature-level fusion experiment using the features of EEG trials and facial video clips with the proposed fusion method was better than unimodal and other fusion methods. The results were also compared with the state-of-the-art methods in the literature to prove the significance of the proposed method.

The existing bimodal emotion recognition methods based on EEG trials and facial video clips are highly complex in terms of computational cost and architecture design [11]. To overcome the issues of these emotion recognition systems, a bimodal emotion recognition method based on the fusion of facial video clips and EEG trials is proposed. The distinctive contributions of the proposed work are summarized as follows:We proposed an efficient and lightweight multimodal emotion recognition model based on two modalities, i.e., EEG trials and facial video clips, by removing irrelevant channels from EEG trials and frames from video clips. In comparison to state-of-the-art approaches, the suggested method’s computational overhead is also low.A video clip contains a large number of redundant frames, which increases the computational overhead of a deep learning model. Selecting the most representative frames helps improve the performance of the method. We proposed a technique to reduce unnecessary frames by calculating the difference between successive frames, organizing the difference frames according to their respective information, and choosing the most discriminative frames from video clips.In addition to wasting time and money by utilizing more electrodes, superfluous channels can impair performance by introducing noise and artifacts into the system. Therefore, in this work, the number of EEG channels used for emotion recognition has been reduced, which ultimately allowed us to design a light weight 1D-CNN model with a small number of learnable parameters. To achieve this, we used pooling layers instead of fully connected layers and depth-wise separable convolution to subtly reduce a large number of parameters and make our network structure simpler for low-dimensional data.We adapted ResNet50 for extracting discriminative information from video clips. Following that, DCCA was designed to fuse highly correlated features from the two modalities. In DCCA, the features from EEG and facial video clips are processed through the two 1D-NNs and then forwarded into a canonical correlation analysis (CCA) layer, which consists of two projections and a CCA loss calculator. While minimizing the CCA loss, highly correlated features are extracted that can be used for the classification. The 1D-NN model was specifically designed to transform the features into a better understanding for correlation analysis while keeping the complexity as low as possible. Extensive experiments were performed to validate the proposed method on two benchmark public datasets.

The rest of the paper is organized as follows: Section 2 covers related work on EEG, facial video clips, and multimodal-based emotion recognition models. Section 3 proposes data preparation, feature extraction methods using CNN, and fusion methods for classification. Experimental results and discussions are presented in Section 4. Finally, the conclusion and future work are discussed in Section 5.

## 2. Related Work

The traditional methods primarily depended on external sources such as facial expressions, body postures, speech, and others for emotion recognition [24]. These sources do not require a subject to wear a set of sensors for procuring these signals, which makes such methods low-cost and less complex. The authors of [25] used facial expressions to recognize emotions. They executed the NN model to identify valence and arousal simultaneously. In light of the accuracy of the method to recognize emotions, the authors suggested that facial expressions are an effective source of human emotion recognition.

Moreover, apart from external sources, internal sources such as EEG, electromyogram (EMG), galvanic skin response (GSR), electrooculogram (EOG), electrocardiogram (ECG), and other physiological signals are also extensively discussed in the literature for emotional recognition [26]. The methods based on physiological signals attain high accuracy in recognizing emotions because these signals can accurately reflect the emotional mood of the subject [27]. According to the different signals discussed, the emotion recognition methods in the literature can be roughly subdivided into three categories: EEG signal-based, facial video clip-based, and multimodal emotion recognition.

### 2.1. Methods Based on EEG Signals

Recently, several investigations have verified the benefits of EEG signals in emotion recognition. In most machine learning problems, researchers’ investigations involve analysis of feature extraction, filtration, and classification tasks. In classification tasks, similar to in every other machine learning problem, EEG data-based classification tasks are also divided into supervised [28] and unsupervised learning algorithms [29]. Supervised learning algorithms need labels with the input data to train the model, such as support vector machines (SVM), K-nearest neighbors (KNN), and others. On the contrary, unsupervised learning algorithms do not need predefined labels and evolve clusters from raw input data on their own, such as K-means clustering, self-organizing maps, etc. 

EEG signals have a high temporal resolution and can demonstrate the association between emotion and brain activity. The equipment used to extract EEG signals is comparatively insubstantial, and the extraction process is also simple [30]. The authors in [28] performed several experiments to fuse graph convolutional neural networks (GCNN) and long-short-term memory (LSTM) neural networks. The model was tested on the DEAP dataset, which resulted in better performance than the existing state-of-the-art methods. The authors in [30] proposed the emotion-dependent critical subnetwork selection algorithm, and the strength, clustering coefficient, and eigenvector centrality of the EEG functional connectivity network features were investigated. The authors in [31] studied a deep, simple recurrent unit network in order to obtain the temporal features from EEG signals, and the experimental results outperformed related work in the literature. It is next to impossible to record a large dataset of EEG signals, but one can explore other methods, such as the cross-subject method. The authors in [32] investigated a unique multisource transfer learning method to detect the emotions of a unique subject using a cross-subject training methodology. This methodology is easy to execute and reduces the need for a large dataset. The authors showed through experimental results that EEG signals can provide useful information about the emotional activity of a person. Anjana et al. [33] converted the time-dependent signal of EEG into scalogram-encoded image data to feed it into a deep-learning model. According to the results, the proposed framework outperformed previous work in the field of emotion recognition using encoded images. Phan et al. [34] presented a unique method to study emotions using EEG signals. The method involved time-domain features mapped into feature-homogeneous matrices. This 3D representation of EEG signals was processed through a 2D-CNN model. The model achieved good accuracy for the valence and arousal binary classification problems. Recently, most of the studies in the literature on EEG-based emotion recognition have relied on deep learning for feature extraction and emotion classification [11]. Moreover, the focus of future research on emotion recognition using EEG is shifting from accuracy to a reduction in complexity [35].

### 2.2. Methods Based on Facial Video Clips

Facial expressions data are one of the key characteristics for detecting human emotions. Numerous advancements have been made in facial expression detection techniques during the past few years. Formerly, feature mining methods included the integral method, the optical flow method, and several machine learning techniques to categorize the expressions. Recently, researchers have switched to deep learning-based methods, such as GoogleNet, for complex image-processing tasks [36]. Deep learning models are extensively used in feature extraction and classification tasks because of their outstanding properties. Moreover, there is no limit to good performance in terms of classification and feature extraction with every unique deep learning-based model, such as SqueezeNet [37,38]. A self-cure network was presented in [39] for expression recognition, and the authors used this novel method to avoid overfitting by efficiently suppressing the uncertainties. The authors in [40] proposed the de-expression residue learning method to recognize facial expressions. This model was capable of learning features from the middle layers of the generative model along with the end layers. Jia et al. [41] studied a label/emotion distribution learning method in an expression recognition problem that used local label correlations in order to minimize the confusion regarding an expression’s description. Recently, approaches that utilize a smaller number of trials are becoming more popular. The one-shot-only method was studied in [42] to upgrade facial expression recognition accuracy and computational cost. Additionally, the authors in [43] studied an efficient structural embedding methodology to reduce the emotional gap between low-level visual features and high-level semantics. Minaee et al. [44] presented an attentional convolutional network to detect facial expressions. The authors used several datasets of facial expressions to show the significance of the suggested architecture. Moreover, the correlation of various emotions with different regions of the face was also reported in the final findings. In an attempt to prove the importance of different emotion recognition models for standard and non-standard facial expressions, Küntzler et al. [45] applied three different facial expression recognition systems. The results revealed the significance of the Azure Face API-based expressions recognition system that performed reasonably well in both standard and non-standard emotion recognition. Although deep learning-based models are becoming more popular day by day, hand-crafted feature extraction techniques are still dominant in the literature [46]. Although hand-crafted feature extraction techniques have proven successful in improving the recognition of expressions, those techniques are time-consuming when it comes to emotion recognition from video clips.

### 2.3. Multimodal Emotion Recognition Methods

Over the past few years, multimodal fusion-based approaches have been presented to facilitate accurate emotion recognition. Multimodal fusion methods acquire signals from different sources to recognize emotion after extracting relevant features from all modalities. The reciprocity amongst various signals and the accessibility of multimodal emotion recognition was proved in [11,26]. Image- and text-based multimodality models were investigated in [47]. Zadeh et al. [48] studied the sentiment analysis method using language, visual, and acoustic modalities, and a tensor fusion network was investigated to fuse data from different modalities. Moreover, there have been several studies that combined physiological signals for emotion recognition. For instance, the authors in [49] proposed EEG and peripheral physiological signals recognize emotions, and the results justified the improvement in accuracy using multiple modalities. Zheng et al. [50] studied a unique multimodal method named the Emotion Meter while using eye movement and EEG signals. The authors developed a deep belief network for the recognition of emotions. Val-Calvo et al. [51] assessed the emotional mood of the subjects while interacting with the HRI system by capturing EEG data, facial expressions, blood volume pressure, and galvanic skin response. Rutter et al. [52] investigated the emotion recognition problem in 644 patients while incorporating self-reported depression severity. Authors reported a decline in emotion recognition accuracy with an increasing age factor in a large number of clinically identified adult subjects with emotional disorders, particularly for negative emotions such as sadness and fear. Aguiñaga et al. [53] used the facial expression as an identifier to extract features from EEG signals and fused both of them to recognize emotions. The author used several classification methods for comparison and proved the importance of one method over the other methods in the literature in a three-class classification problem. In order to detect hidden emotions, Song and Kim [54] designed CNN models to identify the emotion. To detect hidden emotions, the method used EEG signals as primary data for emotion recognition and compared them to relevant facial data. The method performed reasonably well in detecting hidden emotions, and the fusion of two modalities resulted in an improvement in the accuracy of emotion recognition. Hassouneh et al. [55] developed a real-time multimodal emotion recognition model using EEG and facial data. The approach involved a CNN model and LSTM accompanied by an optical flow algorithm for virtual facial markers to recognize emotions by fusing the features of two modalities. The reported results revealed that the algorithm performed well on the personal dataset. However, the method was not tested on any other publicly available datasets for comparison. While considering the importance of time complexity in feature extraction and model training, Lu et al. [56] improved the VGG-Face network model for facial expressions and the LSTM for EEG data. Moreover, the method involved a decision-level fusion method for a multimodal emotion recognition model. Not only did the proposed method outperform the old LSTM model in a six-class classification problem in terms of emotion recognition accuracy, but it also improved the running time of the emotion recognition model. Zhao and Chen [57] proposed a unique method of multimodal emotion recognition. The method consisted of a bilinear convolution network (BCN) to extract features from facial data, and then EEG data were transformed into three frequency bands to feed them into the BCN model. Further, an LSTM-based fusion model was designed for the fusion of features from the two modalities. The model showed improvement in terms of accuracy compared to other methods in a two-class classification model.

EEG-based models show that the performance of the multiple-feature selection procedure is better than the univariate method [13]. Furthermore, the rate of emotion recognition from EEG features extracted with deep learning models was found to be higher than with traditional methods.

The methods discussed in the literature confirmed the effectiveness of EEG and facial video clips for emotion recognition. However, EEG data are comparatively sensitive and becomes attenuated due to low-quality electrodes. The subject’s facial expressions are insufficient for a fair judgment if the subject’s internal emotional state is different from their expressions. Meanwhile, pure external performance is only part of expressing emotion and cannot show the rich emotions of humans. The physiological variations are influenced by the nervous system of the body, which can more accurately depict the emotional mood of the person. Consequently, the fusion of the physiological and non-physiological signals for the recognition of emotion is a unique research trend among researchers around the globe. Facial video clips and EEG signals have been widely studied in a non-physiological and physiological framework and can be efficiently fused for bimodal emotion recognition. Therefore, this collaborative association allows the corresponding information to enhance the objectivity and accuracy of emotion recognition.

## 3. Materials and Methods

In this section, we discuss the datasets, their pre-processing, and the specifics of the proposed methodology.

### 3.1. Emotions

Scherer [58] presented a definition of emotion in 2005, stating that organicistic reactions of the human nervous system to certain events cause emotions in human beings. Moreover, Ekman et al. [59] and Russell [60] resented theories for identifying and categorizing emotional stimuli. Ekman et al. proposed that regardless of the environment and background, certain emotions are inevitable in every human being, namely: happy, anger, sad, fear, disgust, and surprise. Russell [60] presented the circumplex model, which contains levels of activation of emotions, and linked those levels of activation in valence and arousal space with the emotional states of humans. We created a three-class emotional model by combining them: happy (high arousal and valence), sad (low arousal and valence), and neutral (mid-range arousal and valence). In this work, we labeled high valence and arousal with SAM rating ≥ 5, low arousal and valence with a SAM rating ≤ 4) and mid-range arousal and valence with SAM rating between 4–5.

### 3.2. Datasets and Pre-Processing

In this work, we executed offline experiments using MAHNOB-HCI and DEAP datasets. 

#### 3.2.1. MAHNOB-HCI Dataset

The MAHNOB-HCI [61] dataset comprises EEG, video, audio, gaze, and peripheral physiological data of 30 subjects. EEG data were collected using 32 active electrodes on a 10–20 international system with a Biosemi Active II system. Facial video clips of subjects were recorded at 60 frames per second, and the resolution of each original frame was 720 × 580 pixels. The facial video clip data were synchronized with the EEG data with a sampling rate of 256 Hz. To create this dataset, each subject was subjected to watching 20 video clips extracted from Hollywood movies and other sources. The duration of stimulant videos ranged between 35–117 s. After watching each stimulant video, the subject was given self-assessment manikins (SAMs) to rate their judged arousal/valence on a discrete scale between 1–9 [13]. Here, we considered only 27 participants for this experiment. 

#### 3.2.2. DEAP Dataset

The DEAP [62] dataset contains the EEG, video, and other peripheral physiological data of 32 subjects. These data were recorded while the subjects were asked to watch 40 one-minute music videos. During recording, the resolution of each video frame was set to 720 × 576 pixels at 50 frames per second. Further, this dataset includes ratings from each subject for each stimulus in terms of levels of arousal/valence. Note that we used only 22 participants for whom both the facial video clips and EEG data were available for all 40 trials.

#### 3.2.3. Data Pre-Processing and Augmentation

In video data containing facial video clips and video sequences that included undesirable backgrounds, we performed some pre-processing steps to remove the undesirable background. Further, the face of the participant in the videos was not centered, which could pose a problem in extracting useful facial features. Therefore, the participant’s face was centered by applying the combined processes of cropping and brightness adjustment in each video. Additionally, the height × width of video sequences was adjusted to 224 × 224 pixels in order to make them conform to our designed CNN model. Originally, the data containing facial video clips consisted of frames with redundant data. Therefore, before giving the data to the CNN model for feature extraction, it is necessary to discard such frames that contain redundant data. We can reduce not only the dimensions of the input data but also extract the frames containing meaningful data for emotion recognition in this manner. For that purpose, for *T* total frames, we performed a difference operation between every two consecutive frames, which resulted in *T* − 1 difference frames containing zeros where the data matched and non-zero where it was unmatched. Then, for each difference frame (*d_t_*), we calculated the average value (*m_t_*) by using the following Equation (1):(1)$$m_{t}=\;\frac{\sum_{x=0}^{M}\sum_{y=0}^{N}d_{t}\left(x,y\right)}{M\times N},$$
where *d_t_* (*x*, *y*) denotes the pixel value of the *t*th difference frame, *M* and *N* denote the length and width of the frame under consideration, respectively.

Afterward, the average values are sorted in descending order to obtain the difference frames with the highest average values. This whole process is depicted in Figure 1. As the value of mt decreases, the meaningful information in the particular frame also decreases. After running multiple experiments, only 40% of the frames per second were considered adequate for emotion recognition, which helped to preserve meaningful information, and the rest were discarded. This entire process contributed to the reduction of 60% of redundant information.

For the DEAP dataset, EEG data were preprocessed at 128 Hz. However, for the MAHNOB-HCI dataset, the EEG data were preprocessed by applying a bandpass filter on the EEG data to keep the band 4–45 Hz; this helped reduce the artifacts of utility frequency and eye gazing. Head-moving noise was removed by spatial filtering using independent component analysis (ICA). Several studies in the literature have proved the importance of the frontal and temporal lobes on emotion recognition using EEG data [14,64]. However, most of the researchers have considered up to 62 electrodes for emotion recognition using EEG [50]. Apart from the unnecessary time and costs of using more electrodes, the extraneous channels can cause noise and artifacts in the systems, which ultimately affect the performance. Therefore, there is a rising trend among researchers to look for alternative ways of using a smaller number of electrodes to extract EEG signals for emotion recognition [11]. In this work, we used five pairs from the frontal and temporal lobes which helped in better recognition of emotion [21,65]. Selected pairs of electrodes are: FP1, FP2, AF3, AF4, F3, F4, F7, F8, T7, and T8. It also aids in reducing the dimension of the input data as well as the computational complexity.

For each trail, we considered 60 s of facial video clips and EEG data in this experiment. For training the CNN, the amount of data required to achieve respectable accuracy was not adequate. To proliferate the data, a window of 5 s was applied, which resulted in 12 samples from each trail of 60 s.

### 3.3. Proposed Method

In the proposed method, the CNN architecture was chosen for feature extraction and emotion recognition tasks. The method consists of two distinct models for different tasks. The first CNN model is designed to extract features from EEG data, while the second CNN model is used for feature extraction from facial video clips. Once the features are extracted, feature-level fusion is introduced with the help of DCCA, and highly correlated features are fed to the SoftMax layer for classification. The proposed model is depicted in Figure 2.

#### 3.3.1. CNN Model for EEG

Some of the common CNN-based deep learning models usually contain a fully connected (FC) layer at the end for feature extraction, and the FC layer comprises a large number of parameters. For instance, VGG Net [66] consists of FC layers at the end of its architecture, which make up around 90% of their parameters. The VGG16 succeeded in improving performance due to the increased depth of the network. Moreover, the improvement of VGG16 over the other models was also due to the use of convolution kernels of size 3 instead of larger convolution kernels. When compared to large convolution kernels, several layers of small convolution kernels perform better because the depth of the network increases due to several nonlinear layers, and it creates more complex patterns in the learning process without increasing the number of parameters. However, VGG Net uses more computing resources and contains a number of parameters, which leads to more memory usage. The first layer of this architecture makes the most of the parameters. Therefore, we performed several experiments to find a suitable lightweight 1D-CNN architecture without compromising accuracy.

A 1D-CNN model is proposed for EEG data. After the input layer of EEG data, we used a temporary layer to convert the data into 1D to be used for the 1D-CNN model. As depicted in Figure 3, Layer 1 is composed of BN (batch normalization) and Conv1D. BN is known for normalizing the output from the previous layer in a CNN model in modern neural networks and for regularizing the data to avoid overfitting. After normalizing the 1D EEG data, the first convolutional layer (Conv1D), with 1 × 3 sized 64 kernels and stride 1, is applied to the normalized data to obtain features. Further, a rectified linear unit (ReLU) activation layer is implemented after the Conv1D layer to activate nonlinearity. The combined mathematical effect of the Conv1D and the ReLU can be defined as follows:(2)$$x_{j}^{k}=\sigma (\sum_{i=1}^{N_{k-1}}Conv1D\left(w_{i,j}^{k},x_{i}^{k-1}\right)+b_{j}^{k}),$$
where the features maps from previous (*k* − 1)th layer is denoted by $x_{i}^{k-1}$; the resulting *k*th layer’s *j*th feature map is represented as $x_{j}^{k}$; while $w_{i,j}^{k}$ symbolizes the convolutional kernel; total number of feature maps in the preceding(*k* − 1)th layer is represented by $N_{k-1}$; *Conv*1D denotes the convolutional operation without zero padding; the bias for the *k*th layer and *j*th feature map is symbolized as $b_{j}^{k}$; the ReLU activation function is signified as σ(). ReLU is defined as follows:(3)$$\sigma \left(x\right)=\left\{\begin{array}{l} 0,\;\;\;\;\;x\;\leq \;0\\ x,\;\;\;\;\;x>0 \end{array}\right.,$$

The features obtained from the Conv1D layer are processed through Layer 2. Except for MaxPooling1D, Layer 2 is similar to Layer 1 in terms of processing complexity. In Maxpooling1D, the mathematical calculations are defined as follows:(4)$$\rho_{j}^{n}=\max(\rho_{j}^{n^{\prime }}:n\;\leq \;n^{\prime }<n+s),$$
where the $n^{\prime }$th neuron in the *j*th feature map without the max-pooling process is denoted by $\rho_{j}^{n^{\prime }}$; resultant *n*th neuron in the *j*th feature map, after the max-pooling process, is denoted by $\rho_{j}^{n}$, and the size of the pooling window is represented by *s*. In this Maxpooling1D, *s* is equal to 2 with stride 2. In the proposed model, the max-pooling process causes the number of trainable parameters to reduce meaningfully, resulting in accelerating the training process. Maxpooling1D is followed by Layer 3, which is similar to the previous layer, but only the convolutional kernels in the Conv1D are set to 32. After the feature maps pass through Layer 3, the obtained feature map is fed to the dropout (0.5) layer to avoid overfitting. Finally, the FC layer is fed with the output feature maps from the preceding layer.

A grid search approach was utilized to find the optimal hyperparameters. The grid search involved several hyperparameters that are listed in Table 1 with their corresponding ranges and the chosen optimal values for the architecture. We tested the influence of these parameters on the performance of the recognition accuracy and chose the parameters that helped in achieving the best accuracy. The final experimental configurations for the investigation of the EEG data in this work are as follows: the learning rate is set to 0.001, the maximum number of iterations is limited to 40, the ReLU function is used for each hidden layer, the SoftMax output is used for classification, and validation is performed through leave one subject out. The data split for training, testing, and validation was performed such that, from the data of *N* subjects, (*N* − 1) subjects*Number of trails for each subject for training (90% training set, 10% validation) and 1 subject*Number of trails for each subject for testing. Moreover, the regulation parameter was set to be 1e5, cross-entropy loss, and a stochastic gradient descent optimizer. Further, we selected 15 optimization steps based on the validation set as early stopping criteria, Xavier initializer as weight initializer, and the bias vector initialized to all zeros.

#### 3.3.2. CNN Model for Facial Video Clips

The CNN architectures are verified and very well known for image recognition in the world of researchers because of their ability to extract discriminative features for better classification [67]. ResNet [67] outperforms all other CNN architectures in classification by increasing the depth of the network to generate features with more relevant characteristics. In this work, ResNet50 is used to extract features from facial video clips for an emotion recognition task. Figure 4 depicts the structure of the proposed CNN model with the ResNet50 network.

For feature extraction, an input layer provides sequences of facial data to CNN after preprocessing. A dimension reduction layer is added before feeding the input to Resnet50 so that the shape of the input data is transformed to 224 × 224 × 3. We removed the final FC layer of ResNet50 and replaced it with one dropout layer (0.5) and a SoftMax output layer of three emotion classes. We used stochastic gradient descent as an optimizer with a learning rate of 0.001 and a batch size of 32. Further, we replaced the average pooling layer before the FC layer in ResNet50 with the max-pooling layer to subsample the input to reduce its size, which helps decrease the calculations performed in subsequent layers. 

#### 3.3.3. Feature Level Fusion Using DCCA

In this work, we used deep canonical correlation analysis (DCCA) to fuse highly correlated features from EEG and facial video clips. Initially, DCCA was presented by Andrew et al. [68] to compute representations of several modalities by processing them through multiple stacked layers of nonlinear transformations. Figure 5 depicts the architecture of DCCA used in this work.

We applied a grid search approach to find optimal hyperparameters for designing the deep learning model to be used in the DCCA method. After a series of time-consuming experiments, we selected the regulation parameter to be 1e5, cross-entropy loss, and a stochastic gradient descent optimizer. Further, we selected 15 optimization steps based on the validation set as early stopping criteria, Xavier initializer as weight initializer, and the bias vector initialized to all zeros. Moreover, Table 2 presents the tunable hyperparameters and the corresponding ranges that have been considered during the grid search approach to find the optimal values. We tested the influence of these parameters on the performance of the recognition accuracy and chose the parameters that helped in achieving the best accuracy.

The 1D model is developed using CNN architecture to improve the DCCA. The model comprises an input layer, three convolutional layers, each attached with a max-pooling layer, followed by one dropout layer, and finally, a SoftMax layer. The convolutional layers are developed using 128, 256 and 512 convolution kernels with kernel sizes of 3 × 1, 5 × 1, and 3 × 1 and stride of 1 in each layer, respectively. The non-linear function ReLU is used as an activation function. Each max-pooling layer is designed with a pool size of 2 and a stride of 2. The dropout layer is designed with parameter 0.4.

In this work, DCCA is used for feature transformation, and then, the transformed features are fused together to apply classification. The DCCA model is shown in Figure 5, in which we applied a deep learning model for feature transformation, where the CCA layer calculates the correlation, and we use that correlation for feature fusion and classification. Let the matrix $I_{1}\in R^{M\times n_{1}}$ contain trials of the EEG modality, and matrix $I_{2}\in R^{M\times n_{2}}$ contains trials of the facial video clip modality. Here, the total number of trials is denoted by *M*, and the dimensions of features in the EEG trials and facial video clips are represented by *n*_1_ and *n*_2_, respectively. To reorganize the input features in a non-linear manner, we designed a deep neural network for each modality as follows:(5)$$\begin{array}{c} O_{1}=f_{1}\left(I_{1};\;H_{1}\right)\\ O_{2}=f_{2}\left(I_{2};\;H_{2}\right), \end{array}$$
where parameters of the nonlinear transformation are denoted as *H*_1_ and *H*_2_; the resulting features from each neural network are denoted as $O_{1}\in R^{M\times n}$ and $O_{2}\in R^{M\times n}$ and the dimension of features from DCCA is denoted as *n*. Mutually learned parameters *H_1_* and *H_2_,* resulting from DCCA, have elevated the correlation between *O*_1_ and *O*_2_ as high as possible:(6)$$\left(H_{1}^{*},H_{2}^{*}\right)=\arg\underset{H_{1},\;H_{2}}{\max}corr\left(f_{1}\left(I_{1};\;H_{1}\right),\;f_{2}\left(I_{2};\;H_{2}\right)\right),$$

Mutually learned parameters *H*_1_ and *H*_2_ were updated using the backpropagation algorithm. The gradients of the objective function were calculated as suggested by Andrew et al. [68] to reach the desired solution. After the training of the two neural networks, the transformed features *O*_1_ and *O*_2_ ∈ *S* are in the joint hyperspace *S*. Primarily in DCCA [54], the authors did not clearly report the usage of transformed features. The user is free to opt for an approach to make use of the transformed features in the best interest of their application. In this work, we obtained fused features from transformed features as follows:(7)$$O=\alpha O_{1}+\beta O_{2},$$
where α and β symbolize weights sustaining α+β=1. The features *O* combined through DCCA are fed to the SoftMax classifier. The classifier is trained to perform emotion recognition tasks. As previously mentioned, there are several advantages to designing DCCA for data fusion from multiple modalities. For instance, at feature-level fusion, DCCA obtains *O*_1_ and *O*_2_ explicitly for each modality to observe the characteristics and correlation of modality-centric transformations. Moreover, the nonlinear mapping functions *f*_1_(·) and *f*_2_(·) can be controlled to preserve emotion-centric information. Further, in the weighted sum fusion, we are using equal weights for both modalities.

#### 3.3.4. Baseline Fusion Methods

We considered several feature-level fusion methods to fuse the features of EEG trials and facial video clips and applied the classification to ensure the validity and competitiveness of the proposed DCCA-based fusion method for emotion recognition. The comparison shows the importance of the DCCA-based fusion approach over other fusion methods. The fusion methods considered for comparison are discussed below. 

(1) Simple Concatenation Fusion [69]: It is a feature-level fusion method. First of all, the feature vectors from each modality are normalized to zero mean and unit variance. Then, the features are combined. For instance, *I*_1_ = $\left[u_{1},\ldots ,u_{k}\right]\in R^{k}$ and *I*_2_ = $\left[v_{1},\ldots ,v_{l}\right]\in R^{l}$ denote the feature vectors from each modality, and the combined features can be calculated as follows:(8)$$O=\left[u_{1},\ldots ,u_{k},v_{1},\ldots ,v_{l}\right],$$

(2) Multiple kernel learning (MKL) [70]: MKL is widespread for its capability of instantly learning kernels and can be utilized for feature-level fusion. For the MKL problem, a suitable method is to consider that *K* (*x*_1_, *x*_2_) is actually a convex combination of basis kernels:(9)$$K\left(x_{1},x_{2}\right)=\sum_{n=1}^{N}w_{n}K_{n}\left(x_{1},x_{2}\right),$$
where *N* denotes the total number of kernels, *w_n_* ≥ 0 and $\sum_{n=1}^{N\;}w_{n}=1$.

(3) MAX Fusion [69]: It determines the resultant class by choosing the class with the highest probability. Therefore, it is a decision-level fusion method. Suppose that there are *U* classifiers and *V* classes; the probability distribution for each trial can be defined as $P_{j}\left(y_{i}|x_{t}\right)\;\mathrm{where}\;j\in \left\{1,\ldots ,U\right\}$ and $i\in \left\{1,\ldots ,V\right\}$, *x_t_* denotes a trial, *y_i_* signifies the resultant class label, and $P_{j}\left(y_{i}|x_{t}\right)$ symbolizes the probability of trial *x_t_* of *i*th class resulting from *j*th classifier. The mathematical formula of MAX fusion can be defined as follows:(10)$$\hat{Y}=\arg\underset{i}{\max}\{\underset{j}{\max}P_{j}\left(y_{i}|x_{t}\right)\},$$

(4) Fuzzy Integral Fusion [69]: It is a decision-level fusion technique. Let a fuzzy measure *λ* on the set *X* is a function: λ : PX→0,  1, which ensures the following the two axioms: (1) *λ* (∅) = 0 and (2) A ⊂ B ⊂ X indicates *λ*(A) ≤ *λ*(B). In this work, we considered the discrete Choquet integral to join the features of the two modalities. The discrete Choquet integral of a function h : X  →  ℝ+ with respect to *λ* is defined by
(11)gh=∑i=1nhxi−hxi−1λAi,
where subscript *i* specifies that indices are permuted as 0 ≤hx1 ≤ ⋯ ≤hxn, Ai=xi,…,xn, and hxo=0. In this work, we utilize the algorithm suggested by Tanaka and Sugeno [71] to calculate the *λ*. This algorithm helps in minimizing the squared error of the model by calculating *λ*. Tanaka and Sugeno [71] also presented that the minimization problem can be resolved through a quadratic programming technique.

(5) Adaptive Fusion [54]: It is a decision rule-based fusion method. The fusion method is accompanied by a deep learning model, as suggested in [54]. Finally, the decision is made using the following equation:(12)$$y=\left\{\begin{array}{ll} y_{F}+\left(1+\left|\frac{y_{E}-0.4}{100}\right|\right)y_{E}\;\;\; & if\;\left|y_{E}\right|>0.4\\ \;\frac{y_{E}+y_{F}}{2} & otherwise \end{array}\right.$$
where $y_{F}$ denotes facial modality and $y_{E}$ denotes EEG modality. If the value of $\left|y_{E}\right|$ is greater than 0.4, EEG modality is given more weight; otherwise, both modalities are given equal weight.

(6) Bidirectional Long-term and Short-term Memory (BLSM) Network [57]: A three-layered bidirectional LSTM network is designed for feature-level fusion. The complete network is designed and implemented as described in [57]. The BLSM network’s first layer performs the modeling of raw features into time series representation. The subsequent layer fuses hidden features of the two modalities using linear functions, while the nonlinearity factor is introduced with a sigmoid function to represent the fused features in a new representation. The last layer of BLSM is responsible for the time series representation of the output from the preceding layer.

## 4. Experimental Results

The proposed architecture was trained over the MAHNOB-HCI and DEAP datasets. This section provides an overview of the results of single modalities and the fusion of EEG and facial video clips in terms of accuracy.

### 4.1. Experimental Setup

For both datasets, a three-class problem (happy, neutral, and sad) is considered for testing the proposed model. A leave-one-subject-out strategy was used to conduct experiments for each dataset, and the results were compared with several state-of-the-art methods. In a leave-one-subject-out experiment, training data comprise all subjects except one subject, which is left for testing. Moreover, the data split for training, testing, and validation was performed such that, from the data of N subjects, (N − 1) subjects × Number of trails for each subject for training (90% training set, 10% validation) and 1 subject × Number of trails for each subject for testing.

### 4.2. EEG-Based Results

The model proposed in Section 3.2.1 was extended with a SoftMax layer at the end for the classification of emotions using EEG data. The proposed model was successfully able to recognize emotions with an average accuracy of 83.27% and 74.57% using the MAHNOB-HCI and DEAP datasets, respectively. For each user, Table 3 shows the test results using MAHNOB-HCI and DEAP datasets. It can be observed in Table 3 that there are fluctuations in accuracy between users within a dataset. This is because the training data did not include any trials from the test data.

### 4.3. Facial Video Clips-Based Results

The proposed model in Section 3.2.2 was extended with a SoftMax layer at the end for the classification of emotions using facial video clips. The proposed model for facial video clips was able to identify emotions with an average accuracy of 92.4% and 90.5% for the MAHNOB-HCI and DEAP datasets, respectively. For each user, Table 4 shows the test results using the MAHNOB-HCI and DEAP datasets. It can be observed in Table 4 that there are fluctuations in accuracy between users within a dataset. This is because the training data did not include any trials from the test data. The average results from facial video clip data are satisfactory.

### 4.4. Results of Fusion of EEG and Facial Video Clips Using DCCA

After collecting features using EEG and facial video clips in respective CNN models, the DCCA method from the proposed architecture was used to fuse the features for the classification of emotions. The fusion model was able to identify emotions with an average accuracy of 93.86% and 91.54% for the MAHNOB-HCI and DEAP datasets, respectively. For each user, Table 5 depicts leave-one-subject-out tests for the MAHNOB-HCI and DEAP datasets. It can be seen in Table 5 that the accuracy after the fusion of modalities has improved by up to 2% from the results of facial video clips.

In this work, the multimodal fusion model did marginally better when compared to facial expression-based emotion recognition. The reason could be based on the fact that facial expression-based recognition has shown considerable improvement through the proposed preprocessing approach, but real-time emotion recognition that is only dependent on facial expressions is highly volatile since participants are able to deceive the system so long as they are skilled at acting out fake emotions on their faces. In this regard, the limitations of facial emotion recognition can be considerably offset by the EEG signals- which cannot be deceived. As a result, the EEG signals and facial expressions are complementary to one another rather than substituting for one another, and the multimodal fusion-based emotion recognition utilizing both signals is, therefore, more reliable than using just one of the two signals.

In order to show the dominance of fusion over a single modality, the results of valence detection are presented in Figure 6. We can see that a happy state is easy to detect in part (a), where the facial variations are easily detectable. However, in part (b), the single modality-based facial expression detection model fails to detect the participant’s happy state due to no particular variations in facial expressions. The fusion model helps detect the true state of motion with the help of the EEG modality. It can also be observed that it is unnecessary to keep all the frames of a video clip when there is no particular change in the expression. The fusion model was able to successfully detect the true emotion state of the participant by extracting highly correlated features between two modalities.

Before explaining the results of emotion recognition, the results of the proposed methodology are also evaluated using accuracy, recall, precision, and f1-score metrics [11]. We applied EEG and facial video clip data to detect three classes of emotions (happy, neutral, and sad) because these are the very basic categories of human emotions. Table 6 shows the results in terms of different evaluation metrics by applying DCCA to fuse the features of EEG trials and facial video clips. It can be observed that MAHNOB-HCI has been outstanding against the DEAP dataset in all metrics of evaluations, and this might be due to the quality of signals obtained from the MAHNOB-HCI dataset. Overall, the proposed technique of feature-level fusion on the features from deep learning methods has shown satisfactory performance.

Table 7 and Table 8 present the performance factors for each emotion; happy, sad, and neutral on DEAP and MAHNOB-HCI datasets, respectively. It presents the performance in terms of accuracy, precision, recall, and F1-score. It can be observed that the proposed method shows consistency in detecting all three classes; however, it can detect a happy state more accurately as compared to the other classes, which is because of our division of SAM ratings into categories. As one can observe in Figure 6b, the happy state can be the most affected class in detection using facial data only because the facial expression can be misclassified by the model as neutral. For that reason, we have squeezed the region of rating for the neutral class (4 < rating < 5) as it can equally affect the sad class classification. This eventually contributed to better recognition of happy and sad classes while having little effect on neutral class recognition accuracy.

### 4.5. Comparison of Results with the State-of-the-Art

In this section, we sum up our results from the previous section on the MAHNOB-HCI and DEAP datasets. Table 9 displays the comparison between the proposed DCCA-based fusion approach and four existing fusion methods in terms of accuracy using the MAHNOB-HCI and DEAP datasets. The simple concatenation-based fusion method proposed by Lu et al. [69] resulted in the least accuracy out of all methods considered in this study. This explains why the fusion method needs to be much more sophisticated than simple concatenation when it comes to signals from modalities. Fuzzy integrals-based fusion [69] of two modalities yields better results than the other baseline methods because it utilizes a special approach to minimize the error between the two modalities while correlating the features. The MAX fusion [69] shows good performance on both datasets and is probably the simplest of all the methods considered in this study, but it can be easily defeated if one modality carries features of a misleading class with a strong probability. Cai et al. [70] applied MKL to fuse audio and visual information for emotion recognition, but it does not seem to be equally good for the fusion of other modalities. Moreover, the adaptive fusion [54] was also implemented for comparison, and the performance was similar to the fuzzy integrals. A feature-level fusion using the BLSM network [57] has shown good performance in fusing the EEG and facial video clip features, but still, its accuracy is lower than the adaptive fusion method. As depicted in Table 9, the DCCA-based fusion method achieves the best accuracy of 93.86% and 91.54% on MAHNOB-HCI and DEAP datasets, respectively. This proves that the deep learning-based fusion method (DCCA) is the most effective in extracting the highly correlated features and a more robust method for emotion recognition in terms of the modalities under consideration.

The advantage of our proposed method is proven by comparing it with state-of-the-art methods, and we ensured that the experimental setup of the methods from the literature considered the corresponding number of emotional categories. A comprehensive summary of results from the proposed method and other methods is shown in Table 10. Wu et al. [72] propose a model for facial expression and EEG data fusion based on hierarchical LSTM. The features were combined at each time frame to identify the important signal at the next time frame until the emotion result was predicted at the last time frame. In [73], a CNN and decision tree-based emotion recognition model was proposed to fuse the multiple modalities by using the probability value of each category. The process uses a mixed-fusion approach on facial expressions, GSR (Galvanic skin response), and EEG data, which results in a maximum of 81.2% and 91.5% accuracy using LUMED-2 (Loughborough University Multimodal Emotion Dataset-2) and DEAP datasets, respectively. Zhang [74] investigated a LIBSVM (library for support vector machines) model for a classification task and adopted a fuzzy logic approach for decision-level fusion of the EEG and facial expression signals. The author was able to improve the average emotion recognition rate to 85.7%. Huang et al. [75] proposed two decision-level fusion approaches based on the enumerate weight rule and an adaptive boosting technique to combine facial expression and EEG data. The fusion results achieved 69.75% accuracy for the valence and 70.00% accuracy for the arousal space in an online experiment. Aguiñaga et al. [53] considered a three-class (happy, angry, and sad) classification problem and used the DEAP dataset to evaluate the performance. An approach including a neural network for facial expression recognition and a 1D-CNN model for EEG data was implemented and achieved a maximum accuracy of 83.33%. Zhao and Chen [57] achieved good performance through the fusion of EEG and facial features while analyzing time series data with the help of an LSTM model. However, the proposed scheme shows the best performance against other methods in the literature.

Even if several qualified research procedures were studied, the majority of closely linked research works selected the same number of emotional states. It can be observed from Figure 7 that our proposed model has performed meaningfully better than other methods in bimodal emotion recognition. The proposed model uses a DCCA-based feature-level fusion method, which can extract highly correlated features in bimodal emotion recognition.

Table 11 shows the comparison of the proposed technique with the most recent state-of-the-art methods reported in [11], and it can be observed that the proposed method achieved competitive performance. However, it is essential to note that our method relies on discarding unnecessary information from the input data.

For the sake of a fair comparison, we analyzed the computational complexity of the proposed lightweight 1D-CNN model against other lightweight models on EEG data. For this purpose, the number of network parameters and average processing time (for the feature extraction part only) are evaluated in Table 12. We see that Shi et al. [76] and Saini et al. [77] consumed fewer parameters than the proposed method but were not reliable in terms of results. This might be due to the lesser number of convolutional layers or poor selection of convolutional filters, and sometimes the dropout layer plays a significant role in reducing the overfitting problem. However, the proposed model not only achieved a significant performance in terms of accuracy over the other models but also utilized fewer parameters. It can be observed from Table 12 that the proposed model has fewer parameters than the other lightweight models with the least effect on performance. This is because the pooling layer does not contain parameters. We replaced the fully connected layers with the pooling layers, which significantly reduced the number of parameters. Several experimental evaluations have shown that replacing the fully connected layers has little effect on the model’s accuracy. Additionally, the proposed method saves approximately 96% of the processing time compared to Cordeiro et al. [78]. Although the methods presented by Shi et al. [76] and Saini et al. [77] consumed less time, they could not meet an adequate level of accuracy as compared to the proposed method. Here we deduce that: (1) models by Shi et al. [76] and Saini et al. [77] produced low accuracy due to their relatively simpler structure and the usage of fewer convolution layers; (2) model by Cordeiro et al. [78] produced the best results; however, it consumed more network parameters and processing time; and (3) proposed model achieved significant performance as compared to the others.

At the same time, we also monitored the utilization of the hardware resource by the proposed method and other lightweight models in the training and testing phases on EEG data. For that purpose, we ran our experiments on Intel (R) Core (TM) i7-7700K CPU @ 4.5 GHz processor, using 32 GB memory and two NVIDIA GTX 1080 GPUs. We recorded the percentage of overall processing capacity and the percentage of physical memory occupied by the proposed and other lightweight models when they were trained/tested. 

Table 13 presents the hardware resources utilized by the proposed and other models during the training and testing phases. Although there are no data in the papers to show the utilization of hardware resources of the models considered for comparison, it can be seen in Table 13 that some models are more complex in terms of parameters as compared to the proposed model. As a result, such models’ utilization of hardware resources must be greater than those with fewer parameters. It can be observed that out of all models considered for comparison, the model proposed by Cordeiro et al. [78] has the highest usage of processing capacity, while that of Qazi et al. [79] has the largest memory usage. The proposed model’s processing capacity utilization is slightly high, reaching above 19% during training, but memory utilization is much lower. In terms of computing overhead, it can be shown that the suggested model performs adequately when compared to alternative lightweight approaches.

## 5. Discussion

As we know, a positive mood helps improve efficiency at work and secures the mental health of a human being. On the other hand, a negative mood causes stress, which ultimately leads to depression if not treated well/on time [81]. Social or physical environments usually entice negative moods in humans. Moreover, relevant psychological and physiological symptoms also begin to rise inside the body along with the negative mood. Normally, these negative symptoms fade away with time if the stress does not last much longer; otherwise, it may lead to severe damage to the body if the symptoms persist. Therefore, it is necessary to distinguish between happy, sad, and neutral emotions.

This study performed feature extraction and fusion using data from the two modalities. We ran simulations on the MAHNOB-HCI and DEAP datasets to see how useful the suggested approach was. Both datasets include ratings from each subject for each stimulus in the form of levels of arousal/valence on a discrete scale between 1–9 [13]. The most fundamental and extensively used characteristics for representing an individual’s emotions are valence and arousal. Valence stands for pleasure or contentment; therefore, if valence scores rise, so too would the amount of satisfaction for any given subject (see Figure 8). However, arousal symbolizes wakefulness; therefore, if the arousal value is low, the subject is asleep, and if the arousal value rises, the subject is awake. We used these ratings to convert them into three classes, namely sad (rating ≤ 4), neutral (4 < rating < 5), and happy (rating ≥ 5).

The term “autism” was first used by Asperger [82] and Kanner [83] to describe an organic disorder with severe behavioral, affective, communication, and social skills impairment. This disorder is characterized by a lack of interest in other people, speech disorders, attention deficits, and compulsive and repetitive behavior. A chronic neurodevelopmental illness called autism spectrum disorder (ASD) causes difficulties in social interaction and interpersonal communication as well as constrained, repetitive behaviors and interests [84]. Several studies have revealed that people with ASD have trouble reading others’ facial expressions [85]. A lack of emotional awareness negatively affects one’s ability to manage their own emotions [86] and makes it more difficult to understand the emotions of others, which makes it challenging to interact socially. People with alexithymia are unable to express their emotions verbally, either because they are uninformed of the feelings that go along with these emotions or because they confuse emotional and physical experiences [87]. Because of this, they struggle to build close, intimate connections with others, comprehend the motives and attitudes of others, and come to ethically sound conclusions that consider the perspectives of others. One of the most significant areas of overlap between alexithymia and ASD is these components of alexithymia, together with communication and social skill deficiencies. 

Communications and social interactions suffer when people find it difficult to read the expressions on others’ faces. However, in this study, we have combined the features from EEG trials and facial video clips to recognize the true emotions of a person. The proposed system can enable such socially impaired people to function properly by identifying the true facial expression of others. Additionally, this research may be expanded to discover strategies for helping socially challenged individuals recognize their own emotions and develop appropriate facial expressions.

Nonverbal cues in human contact, such as voice tone/pitch/cadence, body movements/kinesthetic information, gestures, and others, that reveal to others one’s intentions, thoughts, and feelings have long played a significant role in human communication. However, the role of each type of communication becomes more important when a bodily/mentally challenged person wants to convey emotions. Kumar and Ponnusamy [88] presented a multimodal technique to identify the emotions of mentally challenged people from facial expressions, voice, and body language. Further, Castellano et al. [89] fused features from facial expressions, body movements, gestures, and speech to identify emotions using a Bayesian classifier. Therefore, combining signals from several non-verbal means of communication along with EEG and/or facial expressions can be worth studying using the proposed method.

This study was conducted to examine the emotions of a human being using more than one modality while preserving the necessary information and neglecting the redundant information in the modalities under consideration. Thus, it is worth discussing the classification results after designing a bimodal emotion recognition model using EEG and facial video clips on separate deep-learning models and fusing the deep-learning features with another deep-learning-based model to develop a highly correlated feature map. The results suggest that deep learning-based emotion recognition models are worth studying, and there is always room for improvement in such models in terms of efficiency. This method consisted of deep learning models for feature extraction from EEG and facial video clip modalities. Afterward, the extracted deep features were fused again in terms of correlation using DCCA. The fused features were used to recognize the three different classes of emotions.

The generalization capability of a model is better judged in a leave-one-subject-out experiment, and the proposed work executes emotion recognition by considering a leave-one-subject-out methodology for testing. The simulations have depicted that the model has attained satisfactory accuracy in bimodal emotion recognition in a leave-one-subject-out environment. Moreover, the testing of the model was performed using single modalities, and the outcomes showed that the proposed model with the feature-level fusion of features using DCCA could attain satisfactory outcomes.

## 6. Conclusions and Future Work

This work presents a unique bimodal emotion recognition model by performing the fusion at the feature level using DCCA. The proposed model relies on a CNN. We assessed the proposed model on the DEAP and MAHNOB-HCI datasets for performance measurement. Initially, we applied pre-processing steps to remove the noise from EEG data and facial video clip data in the DEAP and MAHNOB-HCI datasets. Moreover, balancing the class distribution helped improve the normalization capacity of the classification model. In the end, leave-one-subject-out testing was performed for all subjects, and the SoftMax classifier was applied to perform classification.

The simulation results proved that the features acquired from the proposed/adapted deep learning models and fused together using DCCA achieved better and more consistent performance. The proposed method helped overcome the missing key features problem and enhanced the performance of emotion recognition from bimodal data by extracting highly correlated features for classification. We proposed an emotion recognition system that was applied to all subjects separately for testing. The model is capable of recognizing the true emotional state, because it does not consider any features from either modality, which can cause ambiguity during classification and enhance the accuracy and constancy of bimodal emotion recognition. This model for bimodal emotion recognition is a powerful interface for brain–computer interaction.

Even if the results obtained from simulations are good enough, cross-subject emotion recognition can be a good way to study this method further. We aim to improve this method by making it robust, optimal, and generalized for multimodal emotion recognition, and by including some emotion-related brain neurogenic analysis in the discussion section. Moreover, it is an established fact that the more data, the better the deep learning model will perform. The two considered datasets (DEAP and MAHNOB-HCI) with a limited number of subjects are not sufficient to fully train deep learning models. Therefore, it would be interesting to study the response of the proposed model with more data generated through new experiments and/or augmentation techniques.

## Figures and Tables

**Figure 1 diagnostics-13-00977-f001:**
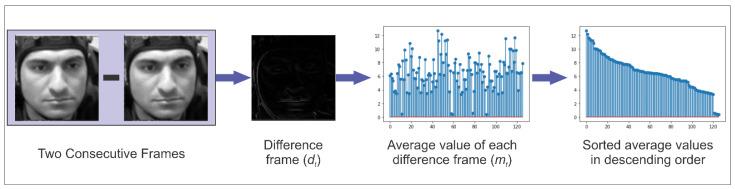
The process of finding frames containing meaningful information (the facial image adopted from [63]).

**Figure 2 diagnostics-13-00977-f002:**
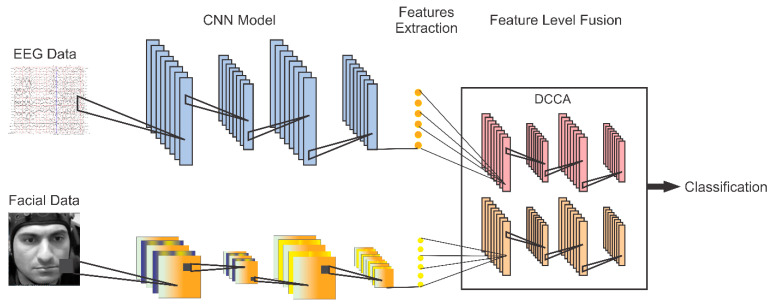
Proposed model for emotion recognition using EEG and facial video clips (the facial image adopted from [63]).

**Figure 3 diagnostics-13-00977-f003:**
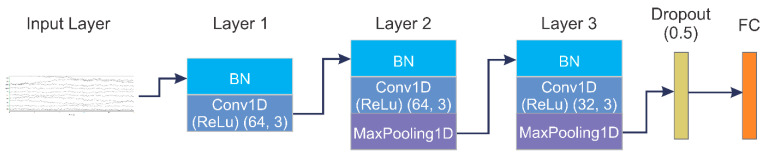
1D-CNN model for EEG data.

**Figure 4 diagnostics-13-00977-f004:**
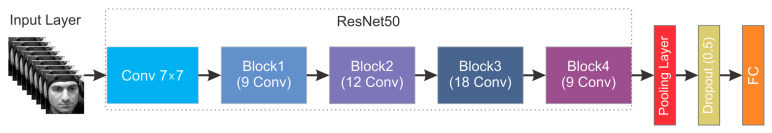
RestNet50-based CNN model for facial video clips (the facial image adopted from [63]).

**Figure 5 diagnostics-13-00977-f005:**
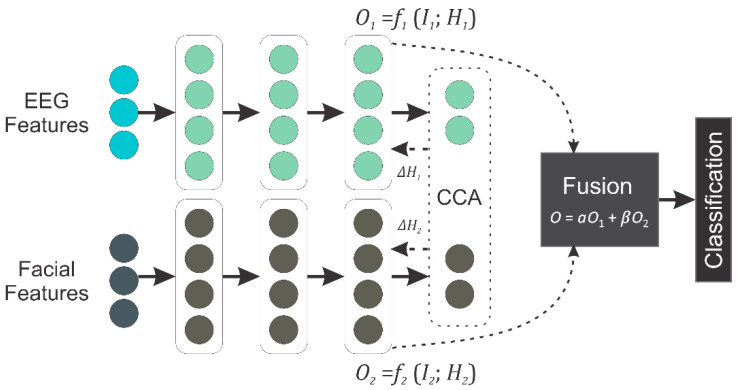
DCCA method used for fusion of EEG and facial video clip features.

**Figure 6 diagnostics-13-00977-f006:**
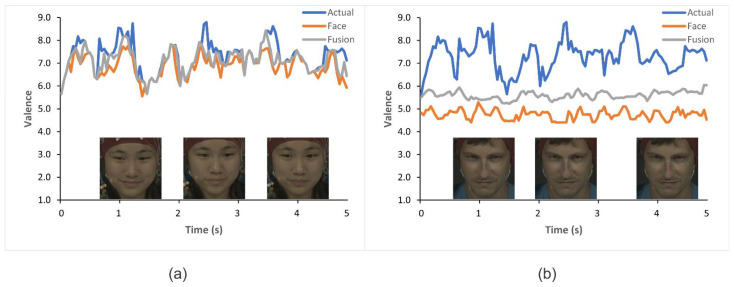
The examples of the valence detection against time are represented for experiment number 10 using the MAHNOB-HCI dataset. (**a**) For participant number 26, the happy state was detected correctly. (**b**) For participant number 1, it was hard to detect any particular changes on the face, but the fusion model detected a happy state successfully.

**Figure 7 diagnostics-13-00977-f007:**
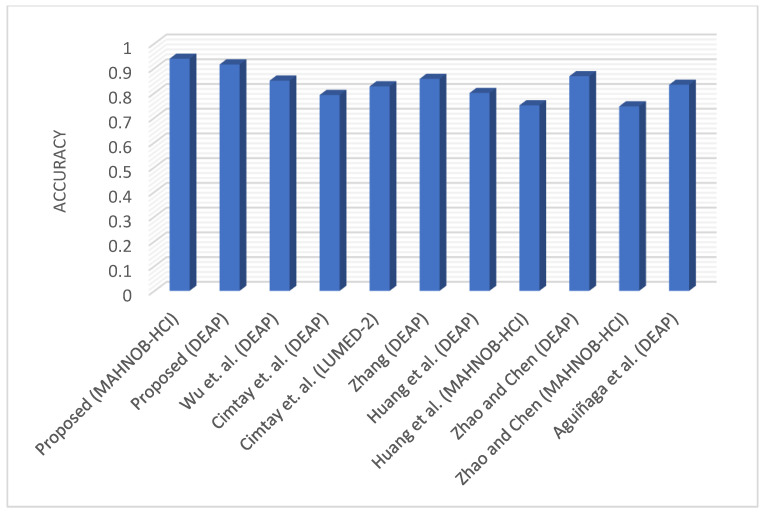
Comparison of proposed method with other state-of-the-art methods from the literature in terms of accuracy (Aguiñaga et al. [53], Zhao and Chen [57], Wu et al. [72], Cimtay et al. [73], Zhang [74], Huang et al. [75]).

**Figure 8 diagnostics-13-00977-f008:**
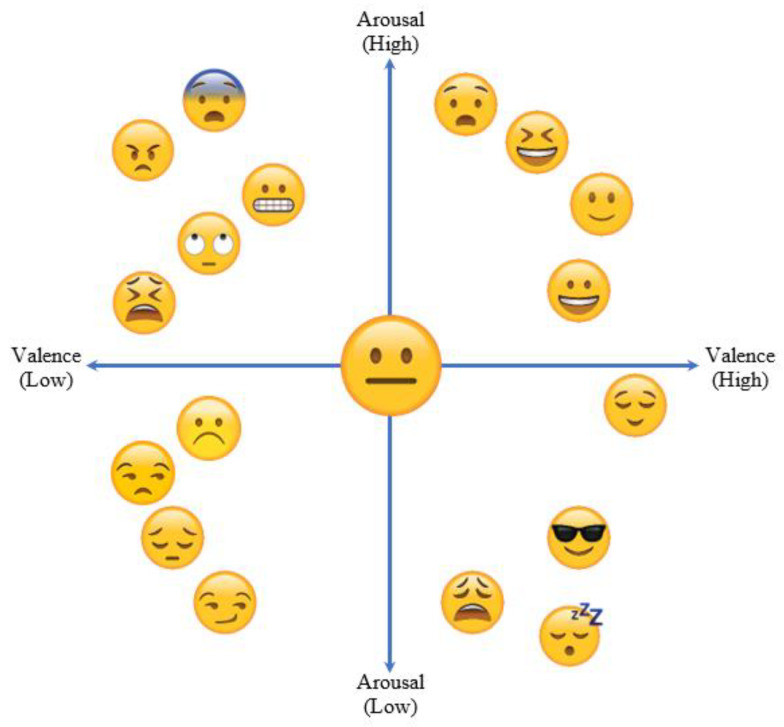
Valence–Arousal chart against various human emotions.

**Table 1 diagnostics-13-00977-t001:** Selected hyperparameters and corresponding range to find the optimal values for a lightweight 1D-CNN model.

Hyperparameter	Range	Optimal Value
Layers	1–5	3
Batch size	18–40	32
Epochs	20–50	40
Learning rate	0.1–0.0001	0.001
Number of convolution filters (Layer 1)	16–100	64
Filter size (Layer 1)	3–5	3
Number of convolution filters (Layer 2)	16–100	64
Filter size (Layer 2)	3–5	3
Number of convolution filters (Layer 3)	16–128	32
Filter size (Layer 3)	3–5	3
Number of convolution filters (Layer 4)	16–128	-
Filter size (Layer 4)	3–5	-
Number of convolution filters (Layer 5)	16–128	-
Filter size (Layer 5)	3–5	-

**Table 2 diagnostics-13-00977-t002:** Selected hyperparameters and corresponding range for the evaluation of optimal values to design a 1D neural network for DCCA.

Hyperparameter	Range	Optimal Value
Layers	1–4	3
Batch size	18–50	32
Epochs	20–60	40
Learning rate	0.1–0.0001	0.001
Number of convolution filters (Layer 1)	100–150	128
Filter size (Layer 1)	3–5	3
Number of convolution filters (Layer 2)	100–300	256
Filter size (Layer 2)	3–5	5
Number of convolution filters (Layer 3)	200–512	512
Filter size (Layer 3)	3–5	3
Number of convolution filters (Layer 4)	200–512	-
Filter size (Layer 4)	3–5	-

**Table 3 diagnostics-13-00977-t003:** EEG-based results in terms of accuracy (%).

Users	MAHNOB Dataset	DEAP Dataset
User 1	89.3	73.7
User 2	84.9	74.5
User 3	78.8	74.1
User 4	95.1	75.6
User 5	83.7	75.9
User 6	96.4	85.4
User 7	79.7	77.5
User 8	76.1	75.3
User 9	90.2	70.9
User 10	84.8	73.6
User 11	83.1	72.5
User 12	86.5	70.8
User 13	94.2	74.9
User 14	92.8	72.8
User 15	92.6	74.2
User 16	83.6	72.1
User 17	72.4	72.5
User 18	71.1	78.3
User 19	72.9	73.1
User 20	87.8	75.2
User 21	74.8	72.8
User 22	71.9	73.3
User 23	78.7	-
User 24	80.3	-
User 25	72.1	-
User 26	85.6	-
User 27	78.8	-
Average	83.27	74.57

**Table 4 diagnostics-13-00977-t004:** Facial video clip based results in terms of accuracy (%).

Users	MAHNOB Dataset	DEAP Dataset
User 1	92.2	88.8
User 2	91.3	88.6
User 3	93.1	90.4
User 4	90.3	89.7
User 5	91.8	89.7
User 6	92.5	91.9
User 7	92.4	91.6
User 8	92.1	93.4
User 9	90.7	87.2
User 10	95.2	87.9
User 11	95.1	91.1
User 12	93.6	92.5
User 13	94.1	92.1
User 14	93.2	90.6
User 15	92.8	92.8
User 16	94.8	87.6
User 17	94.7	90.0
User 18	94.0	91.6
User 19	90.8	91.7
User 20	92.8	91.3
User 21	95.1	90.4
User 22	93.2	91.6
User 23	92.8	-
User 24	93.7	-
User 25	95.8	-
User 26	90.5	-
User 27	93.1	-
Average	92.4	90.5

**Table 5 diagnostics-13-00977-t005:** Results of fusion of EEG and facial video clips using DCCA in terms of accuracy (%).

Users	MAHNOB Dataset	DEAP Dataset
User 1	94.91	89.21
User 2	93.63	91.97
User 3	95.43	90.92
User 4	92.6	92.07
User 5	92.12	90.24
User 6	92.91	92.39
User 7	92.72	91.96
User 8	92.39	93.86
User 9	91.68	89.65
User 10	95.56	90.86
User 11	95.45	91.52
User 12	94.01	92.99
User 13	94.42	92.61
User 14	93.65	91.04
User 15	93.18	93.32
User 16	95.21	89.92
User 17	95.94	90.57
User 18	94.34	92.01
User 19	91.14	92.15
User 20	93.25	91.73
User 21	95.42	90.84
User 22	93.66	92.09
User 23	93.16	-
User 24	94.98	-
User 25	96.19	-
User 26	92.88	-
User 27	93.45	-
Average	93.86	91.54

**Table 6 diagnostics-13-00977-t006:** Results of fusion of EEG and facial video clips using DCCA.

	Accuracy (%)	Recall (%)	Precision (%)	F1-Score
MAHNOB-HCI	93.86	92.24	92.70	0.9351
DEAP	91.54	90.82	90.80	0.9107

**Table 7 diagnostics-13-00977-t007:** Class-wise performance of the DCCA-based fusion method on DEAP dataset.

	Accuracy (%)	Recall (%)	Precision (%)	F1-Score
Happy	93.4	91.1	91.4	0.9127
Sad	90.52	91.3	90.1	0.9004
Neutral	90.7	90.06	90.9	0.919

**Table 8 diagnostics-13-00977-t008:** Class-wise performance of the DCCA-based fusion method on MAHNOB-HCI dataset.

	Accuracy (%)	Recall (%)	Precision (%)	F1-Score
Happy	94.2	93	93.1	0.9413
Sad	94.1	92.32	92.4	0.936
Neutral	93.28	91.4	92.6	0.928

**Table 9 diagnostics-13-00977-t009:** Comparison of average accuracies (%) of different fusion methods with DCCA using MAHNOB-HCI and DEAP datasets.

Fusion Methods	MAHNOB-HCI	DEAP
Simple Concatenation [69]	88.32	85.71
MKL [70]	88.81	86.55
MAX [69]	89.47	87.15
Fuzzy Integral [69]	90.83	88.08
Adaptive Fusion [54]	90.92	88.69
BLSM [57]	88.79	88.14
DCCA [68]	93.86	91.54

**Table 10 diagnostics-13-00977-t010:** Comparison of proposed method with other methods from literature in terms of datasets and number of emotion categories.

Reference	No. of Emotions	Dataset	Accuracy (%)
Proposed	3	MAHNOB-HCI DEAP	93.86 91.54
Wu et al. [72]	2	DEAP	85
Cimtay et al. [73]	2	DEAP LUMED-2	79.2 82.7
Zhang [74]	4	DEAP	85.71
Huang et al. [75]	2	DEAP MAHNOB-HCI	80 75
Aguiñaga et al. [53]	3	DEAP	83.33
Zhao and Chen [57]	2	DEAP MAHNOB-HCI	86.8 74.6

**Table 11 diagnostics-13-00977-t011:** Comparison of state-of-the-art methods in terms of accuracy reported in [11].

Method	Accuracy (%)
LIBSVM	85.71
Spiking neural networks	73.15
CNN	69.38
SVM	73.69
SVM, 3 Nearest Neighbors	66.28
Random Forest, SVM, logistic regression (LR)	73.08
LSTM-CNN	93.13
1D-CNN and neural networks	89.60
LSTM	91.00
Proposed	93.86

**Table 12 diagnostics-13-00977-t012:** Comparison of performance between proposed 1D-CNN model for EEG signals and other lightweight methods.

Methods	Network Parameters	Processing Time (s)	Accuracy on DEAP Dataset (%)	Accuracy on MAHNOB-HCI Dataset (%)
Proposed 1D-CNN model for EEG	19,042	3.823	74.57	83.27
Shi et al. [76]	1482	1.142	68.26	78.76
Saini et al. [77]	2688	2.257	70.13	79.98
Cordeiro et al. [78]	26,224,952	113.484	75.79	85.52
Qazi et al. [79]	1,028,516	59.711	75.08	84.63
Anvarjon et al. [80]	5,137,260	22.231	75.74	84.72

**Table 13 diagnostics-13-00977-t013:** Comparison of hardware resource utilization of the proposed 1D-CNN model for EEG signals and other lightweight methods.

Methods	Overall Processing Capacity Usage Test/Train (%)	Memory Test/Train (%)	Accuracy on DEAP Dataset (%)	Accuracy on MAHNOB-HCI Dataset (%)
Proposed 1D-CNN model for EEG	19.5/20.5	10.9/8.7	74.57	83.27
Shi et al. [76]	17.3/19.67	8.3/7.9	68.26	78.76
Saini et al. [77]	18.5/21.8	9.6/9.0	70.13	79.98
Cordeiro et al. [78]	53.4/65.2	31.9/30.1	75.79	85.52
Qazi et al. [79]	31.49/30.9	24.1/21.4	75.08	84.63
Anvarjon et al. [80]	31.8/29.0	32.5/29/2	75.74	84.72

## Data Availability

The data are available from the corresponding author.

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
