# Peer review of "A Bimodal Emotion Recognition Approach through the Fusion of Electroencephalography and Facial Sequences"

_diagnostics, 2023, doi:10.3390/diagnostics13050977_

Round 1

Reviewer 1 Report

Overall I think this is a good paper. I enjoyed reading this paper. But I think the authors can make this paper even better. For this, it would be nice if the authors are allowed to clarify several points listed below.

What is the contribution of this study?

Andrew et al. (2013) introduced deep canonical correlation analysis, and others (e.g., Liu et al., 2019) applied DCCA for multimodal data fusion for emotion recognition.

Then, exactly what is the contribution of your study? How did you extend Liu et al., 2019 and Andrew et al., 2013? Can you articulate a little more about the contribution (novelty) of your approach?

The title of the paper says “A Bimodal Emotion Recognition Approach through the Fusion of 2 Electroencephalography and Facial Sequences”. Abstract also states: “a deep canonical correlation analysis (DCCA) based multimodal emotion recognition method is presented through the fusion of Electroencephalography (EEG) and facial video clips”. From the title and the abstract, I expect that the main theme and contribution of this paper is the data fusion of EEG and facial video clips using DCCA. Then the contribution part (Line 124-136) suggests little about DCCA. Their contributions are: they used (1) a small number of relevant EEG channels and video frames, (2) lightweight 1D-CNN and again (3) a method to remove the redundant frames and select the most discriminative frames from videos clips. Not much suggested in the contribution section regarding your work concerning DCCA even though the main text is mostly about DCCA and its performance comparison with other data fusion methods. I wonder why. Is this because the current work is one way or another a direct extension of Liu et al. 2019? For that, you chose to de-emphasize your DCCA part?

Tables 4 and 5.

DCCA is only marginally better than single modality face-alone classification (face alone: average (MAHNOB, DEAP) = (92.4, 90.5); DCCA fusion (MAHNOB, DEAP) = (93.86, 91.5) – I suspect they are not statistically different. It is difficult to tell exactly whether DCCA as data fusion method was effective because classification performance using single modality (face alone) was nearly as accurate as fused classification. The problem here is that the two data sets (MAHNOB and DEAP) weren’t good to test data fusion methods because one modality using face video clips was already high (92.4, 90.5) as compared to the other modality (EEG alone 83.27, 74.57). Or maybe I misread Tables 3 and 4.

I am still confused about Table 3 and Table 4. Can you explain a bit more about Tables 3 and 4? Here the single modality means O1 (Table 3) or O2 alone (after DCCA) and multimodality means O = alpha O1 + betaO2? If that is the case, I think the result is interesting because it shows that even with single modality, somewhat incorporating with other modality (selecting projection matrix H1* that are highly correlated with facial video clips) helps emotion recognition.

Table 9

The authors compare their DCCA method to other fusion methods (Table 9); it shows that their DCCA method outperformed other data fusion methods. But isn’t it simply due to the fact that DCCA, as the authors implemented it, has a weighing function O = alpha * O1 + beta * O2; (alpha + beta = 1). Basically doesn’t this show that a particular data set (face over EEG) was effective? In other words, the comparison with other fusion method (Table 11) simply indicates that this weighing method, but not necessarily DCCA per se, worked well. It appears that in DEAP and MAHNOB-HCI, EEG data do not contribute much as compared to visual face clips (Tables 3, 4, 5). For the fused DCCA, EEG features were simply discounted (or nearly discarded) but other data fusion methods didn’t have this weighing function?

Discussion section

The Discussion section consists of some implications and summary but it also contain comparisons with other fusion methods (Tables 9-12), I think these analyses should be included in the Result section as a separate subsection.

Reviewer 2 Report

This paper uses a unique bimodal emotion recognition model relying on a CNN to classify emotions to help enhance human computer interfaces. Its results are very encouraging and its application to a wide range of clinical and extra-clinical settings is strong.

Overall, the methods are scientifically sound, detailed, and well-presented.

Throughout, the English should be edited.

“In many cases, it requires coordinating its operations with the respondents, therefore a framework with emotional intelligence can better adjust in such an environment” should be “In many cases, it requires coordinating its operations with the respondents. Therefore, a framework with emotional intelligence can better adjust in such an environment”.

As another example, “CNN network” should be just “CNN”.

Some food for thought—the answers to which could strengthen the ramifications of your model.

Introduction:

Might it make sense to speak to disorders of emotion recognition to frame the clinical importance of your work (e.g. alexithymia, bipolar disorder, eating disorders, borderline personality disorder, among others)?

Methods:

Might it make sense in the Methods or Discussion to speak to how you might expand your model towards a greater degree of granularity in terms of the emotion recognition task? For example, such that it could identify 30+ emotions, or, better yet, identify someone’s feelings on a multidimensional plot representing the main different “elements” of emotion (e.g. x axis happy-sad, y axis angry-sullen, z axis…)?

Discussion:

·         The methods are impressive and well detailed, but how can your emotion classifier be applied to improve the human condition in health and disease? For example, autism populations have great difficulty recognizing and responding appropriately to emotions (this is known as alexithymia; PMID: 30065681). However, many new technological interfaces have boosted their ability to adaptively engage with the world. Could your classifier be used to teach such individuals how to recognize others’ and their own emotions, communicate them better, and respond appropriately? For lay readers with a background in developmental neurology, such a brief discussion (2-3 paragraphs) would be of immense relevance.

In addition, might your classifier be able to be applied to other sensory modalities for expressing emotions (e.g. voice tone/pitch/cadence, body movements/kinesthetic information)? This would be of great relevance to disabled populations as well—and, evolutionarily, of interest in light of how these different modalities have evolved in humans and across the world. This is more esoteric, but even one sentence speaking to this could strengthen the range of your work’s applications.

Have you explored the different performances of neuro-atypical individuals? For example, one might expect that individuals with bipolar disorder (I or II), who are more sensitive and emotionally attuned, might outperform other individuals. This would be of relevance as a contributing prognostic test, clinically or extra-clinically.

Round 2

Reviewer 1 Report

In my earlier review, I raised the following question:

Tables 4 and 5.

DCCA is only marginally better than single modality face-alone classification (face alone: average (MAHNOB, DEAP) = (92.4, 90.5), Table 4; DCCA fusion (MAHNOB, DEAP) = (93.86, 91.5), Table 5. I suspect they are not statistically different. It is difficult to tell exactly whether DCCA as data fusion method was effective because classification performance using single modality (face alone) was nearly as accurate as fused classification. The problem here is that the two data sets (MAHNOB and DEAP) weren’t good to test data fusion methods because one modality using face video clips was already high (92.4, 90.5) Table 4, as compared to the other modality (EEG alone 83.27, 74.57) Table 3.

I would like to see the authors respond to this inquiry.
